# How Can We Frame Energy Communities' Organisational Models? Insights from the Research 'Community Energy Map' in the Italian Context

Lorenzo De Vidovich [1,*], Luca Tricarico [2] and Matteo Zulianello [3]

1   Department of Political and Social Sciences, University of Trieste, Piazzale Europa 1, 34127 Trieste, Italy
2   Istituto di Ricerca sulla Crescita Sostenibile, IRCRES CNR, Via dei Taurini 19, 00185 Rome, Italy
3   RSE S.p.A, Ricerca Sistema Energetico, Via R. Rubattino 54, 20134 Milan, Italy
*   Correspondence: lorenzoraimondo.devidovich@dispes.units.it

**Abstract:** According to the early transposition of the EU directives by the Italian government, this paper presents some of the outcomes of the qualitative-led applied research titled *Community Energy Map*, aimed at identifying the main operational models and organisational frameworks put in place for the development of renewable energy communities (RECs). In this respect, the article discusses a threefold subdivision of organisational models to implement RECs: public lead, pluralist, and community energy builders' model. Furthermore, the paper illustrates in detail three of the nine case studies dedicated to recently launched RECs, conducted through qualitative fieldworks, to investigate the social and local implications generated by these community-led initiatives. The article stresses the relevance of both the local scale and community-led initiatives in the pathway towards a fair and just energy transition, by discussing how RECs define new organisational models of distributed energy systems.

**Keywords:** renewable energy communities; energy community; sustainability; open sustainable innovation; sustainable energy; Italy

## 1. Introduction

In Europe and Italy, as well as elsewhere in the world, the energy transition from fossil fuels to renewable sources has become a major focus of public debate and a priority for the climate agenda, EU energy, and individual member states. Many crowds, with different backgrounds, roles, and know-hows, are involved in the energy transition process.

The threats behind climate change, which call for an energy transition, are even more heartfelt regarding the choice of models that will represent the future of the European energy system, together with other key debates, such as that on the taxonomic definition of renewable energy sources that enable the EU to enact the transition process according to the European Green Deal (European Green Deal: https://ec.europa.eu/info/strategy/priorities-2019-2024/european-green-deal_en, accessed on 20 December 2022). Within this Special Issue, this paper contributes with a focus on the organisational innovation of community-led processes relating to energy systems, which involves different collective actors to develop energy transition projects, today commonly known as 'renewable energy communities' (hereinafter, RECs).

As this debate arises, it will induce future tension and will affect the relationship between the global and local dimensions of the energy transition. On such a basis, the need to develop a complete organisational analysis framework of these community-based initiatives, as presented in this article, highlights a pivotal issue for the management of energy systems: the need to combine the model of large and centralised energy production in the hands of a few—which is necessary to foster the strategic industrial sector of the country and the balance of the network—with models of distributed generation, oriented towards

energy efficiency and the electrification of energy loads, where possible. These observations are to be contextualised in a broad understanding of the concept of sustainability, by being aware that environmental, social, and governance factors related to such projects play an important role for public decision-makers, companies, and investors alike [1].

Although aimed at sustainability objectives, the previous periods of energy policies have often promoted initiatives and projects oriented towards the extraction of economic value by large national and international operators in comparison to local communities. Against this backdrop, the de-territorialised energy production models are far from the vocations of the territories and, in many cases, compete with the raising of opposition movements by local communities.

It is therefore important to focus on the role of organisational development processes of sharing and aggregation of resources and knowledge. These processes are outlined under the umbrella of local innovation and are facilitated through opportunities arising from local and national policy frameworks. The diffusion of these highly innovative collective models may indeed be subject to the economic resources and assets available to the applicant entities. In the diffusion of these practices, it therefore appears crucial to be aware of the risk of creating new asymmetries and exacerbating further territorial inequalities.

Decentralisation and territorialisation of energy production are the main features on which energy communities are based, insofar as the localisation of energy production towards the aim of making it and energy consumption closer to each other is a pillar of energy community initiatives [2]. This paper aims to analyse how these aspects hold together in the management of local governance of RECs' initiatives, with reference to the involvement of citizens, the development of business activities, the power generation from renewable sources, and the willingness to consume and exchange energy locally.

Localised forms and initiatives of energy communities began to appear in Italy in the Alpine areas at the end of the 1800s, stemming from the valorisation of local resources to satisfy the emerging energy needs of the local communities [3]. The relevance and the choice of delving into the Italian context is therefore linked to the history of these initiatives, which over the years has witnessed an evolution of very different organisational types, but has been associated with the same objectives of international experiences of energy communities [4], in addition to a heated debate on the framework of rules and policies that will support future developments [5].

From the outcomes of these prodromal experiences, that suggest a complex combination between social arrangements and technical issues for the development of RECs' initiatives [6], to research questions that arise, by involving an analysis of both organisational models and social impacts of an REC, by also relating to forms of citizen involvement for the development of energy communities:

(1) What 'organisational lenses' can we adopt to observe renewable energy community projects in the Italian context?

The aim is to systemise tools for analysing the processes by which energy communities and community energy projects are built, identifying the relationship between development phases and the legal profiles that are adopted, while observing the relationships between the different actors that lead the management of these projects. Community involvement means much more than placing smaller renewable energy infrastructures close to consumers. The identification of three 'organisational clusters' enables the distinguishing of three different models about the ways in which community energy initiatives have been framed and developed in the Italian context. The development of a community-based enterprise is based not only on the investment in a specific generation technology, but also, critically, on a process of social and organisational innovation.

(2) Considering the different organisational profiles, is it possible to define models that enable the outline of specific forms of involvement and the social objectives underlying such initiatives?

The aim is to identify rules and modes of engagement, types of actors involved, and relations with third parties (including industry) not directly involved in renewable

energy community projects, but they can nevertheless contribute to the achievement of the objectives identified by the partners or members.

In order to answer these questions and pursue the aforementioned objectives, the article is divided into different sections. The first section provides a theoretical configuration of the field of investigation, thus laying the conceptual and analytical basis for understanding the research activities. The section is completed by a description of the Italian regulatory framework for RECs' development, according to the intense regulatory and legislative activity undertaken between 2020 and 2021.

The second section outlines the research steps, the research methods, and the adopted research tools to perform the empirical and exploratory activities. Research steps are identified as follows: desk analysis about RECs' projects; identification and delineation of three 'analytical clusters' aimed at systemising the RECs' development models; conduct of three focus groups; and in-depth activities through nine case studies. Then, delving into the specifics of the empirical explorations, the third section of the paper studies three particularly paradigmatic and representative examples of the three models identified, divided into a public lead model, pluralistic model, and community energy builders. The application of these three models is exemplified by a brief illustration of three cases. Lastly, returning to a general level of analysis, the paper addresses and discusses the main themes that emerged from the research activities, identifying some still open research and development trajectories. In this sense, the concluding section attempts to describe the social and territorial impacts of the analysed energy communities, the implications (and related innovations) in terms of organisational competences, and the institutional impacts, understood as outputs useful for maintaining a pluralistic and shared ecosystem for a fruitful development of renewable energy communities.

## 2. Defining the Organisational Field of Investigation: Renewable Energy Communities

The notion of 'energy community' is increasingly used as a collective term for all activities that deal with the generation of energy in a community setting [7]. On such a basis, in the field of study related to energy, the term 'community' has often been understood as a specific type of social relations characterised by participatory governance and distributive justice [8].

Many member states in the EU have expectations of a more decentralised and democratic system which would benefit from renewable energy production and move away from passive consumers towards a more dynamic relationship where active energy citizens are engaged and take responsibility for energy production and consumption [9,10]. Transformations in the energy system, combined with technological changes, have revealed an organisational field [11] of new interests and new opportunities for the co-production and supply of energy. To that effect, the transition lends itself to various social analyses, in particular, on account of the fact that it represents a dynamic of social change where technical innovations and social change feed each other [12]. To date, it might be argued that the co-supply of energy products and services involves various forms of cooperation between public institutions and citizens, but also between private multi-utilities, local enterprises, and consumers [13].

New approaches have devoted particular attention to the inclusion and involvement of users and inhabitants, enabling them to collectively develop and manage energy services through a model of co-ownership that differs from the traditional entrepreneurial organisations of energy supply [14–18]. In recent years, the topic of energy community has gained great resonance in the European academic debate [19–24], and in public decision-making arenas [25–28]. Furthermore, the community-based initiatives may also 'nudge' and stimulate citizens' awareness on environmentally related issues [29].

Many European countries have turned towards the study of innovative models of co-operation in the co-production of energy from renewable sources, with positive experiences in Spain [30], Belgium [31], Germany [14], the UK [16,32,33], and the Netherlands [34]. Simply put, these initiatives refer to a pathway where people adopt 'sustainable energy

technologies and strategies in groups and/or on shared ownership, as opposed to the traditional individualistic adoption' [35]. Among these various approaches, Europe and Italy are experimenting with various innovative and sustainable energy models, capable of pursuing the goals dictated by the energy transition through new community-based approaches. After having taken shape in the Anglo-Saxon context [19,32], and driven by recent scientific research contributions able to offer an overview at a European level [36,37], the field of 'energy communities' has progressively gained relevance in Italy, through numerous studies aimed at systematising the research theme [38–42]. At the same time, an intensive reporting activity conveyed the topic to *policy-makers* [43–46]. Within this framework, what has taken shape is the assumption that the success of contemporary energy transition pathways will only be possible through the inclusion and the support of citizenship [47]. In this respect, 'energy community' initiatives have been identified as 'an emergent phenomenon that in the current phase provides a useful grassroots approach for many citizens to engage in the transition to a sustainable energy future' [20].

According to many scholars, these initiatives represent an emerging phenomenon which is generating new opportunities for citizens and local coalitions by offering them the chance to actively participate in the energy market [47], not only as consumers, but also by deciding the form and scope of energy production. Energy communities (RECs) are considered a model of organisational innovation that makes end-customers protagonists of the energy transition, allowing citizens, administrations, and local enterprises to collectively develop and manage energy projects or services, with a different model of governance and ownership with respect to traditional business organisations.

In their pivotal work, Walker and Devine-Wright [19] identified two key dimensions related to the views of those involved or concerned with community energy projects and the relevant policies, and that play a crucial role for our analysis. The first, a 'process' dimension, relates to the study of the organisation and management of a specific community project and to the involvement of inhabitants in decision-making. The second, an 'outcome' dimension, refers to the social and spatial distribution of benefits (i.e., by literally replying to the question 'a project for whom?').

### 2.1. Policy Framework

The socio-technical organisational concept of energy communities has been incorporated by a recent development in the European energy market regulation concerning the relationship between market and consumers [48]. The innovations for the development of energy community projects have recently been regulated by the Clean Energy for all Europeans Package and two directives contained therein: Directive 2018/2001, known as the 'Renewables Directive' RED II, and Directive 2019/944, known as the 'Internal Electricity Market Directive' IEM. These directives define the legislative and regulatory framework for citizen participation, mobilising private capital, and ensuring local acceptance of new experimental renewable energy initiatives [31,49].

According to the definition adopted in the EU RED II Directive 2018/2001, the term 'renewable energy community' (REC) refers to a coalition of users who, through the constitution of a legal entity, decide to cooperate with the aim of fulfilling the needs identified by their members, to produce, consume, and manage energy through one or more local plants powered by renewable sources. The enlargement of the number of actors, contributing to the global objective of decarbonisation and the realisation of an energy transition pathway, cannot disregard the diffusion of new, sustainable (and virtuous) approaches based on communities and territorial realities, as well as the active involvement of citizens and consumers.

In line with the indications provided by the two European directives, which are considered the founding elements of the Climate and Energy Package approved in May 2019 by the Council and the European Parliament, renewable energy communities pursue the fulfilment of environmental, economic, and social needs identified by their members, which are put at the forefront respect for profit-making purposes. The production of energy

from renewable sources should therefore not be understood as a mere economic or financial phenomenon, but rather as a tool for fostering social relations and the generation of tangible spin-offs in the organisation of the local structures that implement and engage the plants.

Within a context of progressive liberalisation of the energy market and decentralisation of energy generation activities, in recent years, the prominence of end-customers has gained in importance, resulting in what Goldthau has defined as a new polycentric governance of the energy infrastructures [50]. They have become prosumers, being consumers who actively participate in the energy production phases as owners of their own renewable energy production plant, of which they consume a part and feed the remainder into the grid, making it available to physically closer consumers, or storing it for consumption at the most appropriate moment.

In pursuit of this objective, the contents of the Clean Energy for all Europeans Package introduced a series of clear guidelines to outline a new phase of innovative experimentation capable of building community forms of access to and consumption of the energy, identified as a commodity of primary necessity around which complex forms of public–private governance take place, especially following the long wave of energy market liberalisation, which has involved a large number of European countries.

### 2.2. The RECs Context in Italy: State of Play

In Italy, the energy transition towards the exploitation of renewable sources is far from being a new phenomenon, although the twofold process of decentralisation and location of energy co-production and consumption has only recently attained political relevance. Indeed, the Italian community energy sector is still a niche market, characterised by small initiatives largely dependent on national photovoltaics (PV) policy support [40]. Although the development of community enterprises in the energy field dates back to the early 1900s—when some prodromic historical experiences of cooperation were developed in the Alpine areas (ARERA, *Registro delle cooperative storiche e dei consorzi storici dotati di rete propria* (Record of historical energy cooperatives and consortia with their own pipelines network: https://www.arera.it/allegati/docs/20/233-20.pdf, accessed on 5 August 2022)—energy communities only began to emerge in Italy from the early 2000s, as a result of several incentives for the deployment of photovoltaic plants. Subsequently, community-led energy initiatives have emerged as a new paradigm capable of increasing the participation of end users (individual citizens, but also public administration and small or medium enterprises) in the international energy transition process.

Even more recently, between March and December 2020, with the aim of launching a trial relating to the introduction of the organisational models proposed by the RED II and IEM directives, all the acts were published. They were necessary to define the legislative and regulatory framework, and the incentive mechanisms to be applied for renewable energy communities and collective self-consumption schemes (Law 8/2020, converted into Article 42/bis DL 162/19-*Decreto Milleproroghe*, ARERA Resolution 318/2020 and the Ministerial Decree of 16 September 2020, by the Italian Minister of Economic Development). The partial and early transposition of the RED II Directive has made it possible for the first time to formally establish renewable energy communities (RECs) in Italy, albeit with the number of constraints related to the maximum power of the individual plants held by the communities and the proximity of the plants to the members' contact points. These first trials were conditioned by at least two elements:

(1) The first concerned the set of technical rules to be accredited and receive the incentive, and were published on the GSE website on 22 December 2020;

(2) The second was the draft legislative decrees for the overall transposition of the RED II and EMI directives that were circulated in August 2021.

The development of 'experimental' RECs has been strongly conditioned on the one hand by the definition of the procedures to have access to regulation and dedicated incentives, and on the other hand by the fact that the constraints introduced in the experimental phase will change during the overall transposition of the directives. Many of the actors that

could participate in the development of these initiatives have therefore decided to wait for the evolution of the legislation that will impact the deployment of RECs.

Law 8/2020 describes a well-defined scope of experimentation, specifying that RECs may develop renewable energy plants with a maximum power of 200 kW, connected to the local low-voltage distribution grid. A further constraint to be met is that the plants held by RECs and members/partners must be underpinned by the same MV/LV secondary transformation substation (or pertaining to the same building or residential building in the case of collective self-consumption schemes). In full coherence of the RED II Directive, the law specifies that the objective of an REC is to provide environmental, economic, or social benefits at a community level to its shareholders or members or to the local areas in which it operates, rather than financial profits. Renewable energy communities are based on 'open and voluntary participation' and are effectively controlled by shareholders or members who are located in the vicinity of the production facilities owned by the renewable energy community. The shareholders or members are natural persons, small and medium enterprises (SMEs), or territorial or local authorities, including municipalities, provided that, for private companies, the participation in the renewable energy community does not constitute their main commercial and/or industrial activity.

ARERA Resolution 318/2020/R/eel of 4 August 2020 defines the regulation model to be applied to the RECs, identifying the benefits that these entities bring to the grid and the tariff components that must consequently not be applied to them. In the end, on 16 September 2020, the Italian Minister of Economic Development introduced the incentive framework for RECs and for collective self-consumption. The energy shared by RECs receives an incentive tariff of EUR 110/MWh, while that shared by participants among collective self-consumption schemes amounts to EUR 100/MWh. We thus move from an incentive that rewards the maximum production of a plant to an incentive that rewards the maximum production of energy that can be consumed at the same time (or rather, time period) by the members of RECs.

In addition to the national legislation, the institutionalisation of energy communities in the Italian context has generated the development of several laws of regional initiative, with the intention of locally promoting the development of RECs that are more linked to territorial peculiarities. The first Italian region to have adopted a regional law regarding the modalities for the establishment of an energy community was the Piedmont region, with the regional law of 3 August 2018 no. 12 'Promotion of the establishment of energy communities'. This law established the purposes, competences, and financial support for the constitution of RECs. In this respect, the coordination of regional laws in light of the national regulatory framework seems crucial to avoid the introduction of different definitions at different institutional levels.

The latest update of the regulatory process dates back to the last months of 2021, through the promulgation of the Legislative Decree 199/2021, which contains new indications on the promotion of the use of energy from renewable sources, increasing the arena of potential stakeholders who can take action to establish an energy community and redefining the maximum power, which increases from 200 kW to 1 MW. The long procedure to formally implement the new measurements lasted for the entire year, 2022, and culminated with new documentation from ARERA, released on 27 December 2022 (*TIAD—Testo Integraato Autoconsumo Diffuso*, within the Resolution 727/2022/R/eel).

### 3. Methodology and Research Methods

The methodology adopted in this paper and in the empirical research activities related to the research *Community Energy Map* is characterised by a qualitative approach. Specifically, qualitative research can be defined as 'a process that draws on the understanding of different methodological traditions of survey to explore a social or human problem. The researcher outlines a complex, holistic picture, analyses words, reports in detail the point of view of informants and conducts the study in a natural setting' [51].

On this basis, and according to the research aims of this paper, qualitative methods enable the addition of knowledge on the timely topic of RECs' development within the energy transition, and to lay down some general key aspects derived from the number of case studies. In this regard, the comparison between cases entails a translation of ideas into action, to provide input for policy change and institutional–spatial design [52]. Such a process of translation and transfer always involves comparison, either between cases or with a framework built on well-known examples and previously researched cases [53]. According to a qualitative research framework, we identify three main methods with which we conduct the empirical investigations. They can be divided in chronological order as follows: desk analysis, adopted to define the analytical clusters aimed at summarising some general models of the *modus operandi*; focus groups; and case studies.

### 3.1. Desk Analysis and Clustering Activities

*Desk analysis* aims to survey and collect information on energy community projects, including pioneering experiences that have now acquired a historical character, acquired experiences over the past two decades, and more recent trials.

Beyond the current regulatory framework, which represents the starting point of an evolutionary path for the emerging distributed community generation sector, it is also interesting to observe some 'historical' projects, de facto energy communities that already represent a disruptive case study and are the bearer of numerous innovations in the energy market. It is a sector that might include, in addition to renewable generation initiatives, those of energy efficiency, intelligent distribution, and district heating, up to the management of storage systems and the spread of e-mobility.

An important reference point in the international field that has thoroughly described the potential of these initiatives lies, hitherto, in the report produced by the European Federation of Energy Cooperatives, REScoop, and the non-governmental organisations Friends of the Earth Europe and Energy Cities [27]. The document provides an analysis of 27 case studies located in the European context and focuses on the analysis of management dynamics in the planning of partnerships, while also providing a series of useful practices with respect to the technologies to be used.

On that basis, the first activity was geared towards providing a preliminary analysis of the fundamental characteristics of energy communities, reviewing 58 cases activated throughout the country which are divided as follows: 23 energy community enterprises that do not comply with the current legislation, started in the years preceding the most recent legislative innovations (Figure 1); 8 energy community builders; and 27 renewable energy communities started in compliance with L. 8/2020 (Figure 2).

In Figure 1, pilot projects launched by RSE in recent years are also mentioned, while, as declared in the caption, not all historical cooperatives in the Alpine Space are cited, with three (*CEG Energia*, *Società Cooperativa Gignod*, *e-werk* in Prato allo Stelvio, and the cooperative society *SECAB*, in Friuli-Venezia Giulia region) indicated. In order to achieve greater knowledge of the cases identified, please refer to the tables in Appendix SA (RECs launched or under construction in light of Law 8/2020) and Appendix SB (Energy enterprises non-compliant with the current normative, including the eight community energy builders).

This first exploratory step relied on a 'secondary' analysis, based on sources, data, and information provided by the main dissemination bodies on energy issues and the knowledge of specific cases by individual members of the research team. Desk analysis included joint initiatives of a historical nature dating back to the beginning of last century, and these type of experiences took shape over the previous decades, using the incentive tools that are available at the time, such as the so-called 'Energy Accounts', and, finally, the recently developed energy communities that are taking shape or have been launched in the past two years, in compliance with Law 8/2020. In this regard, as mentioned, Figure 2 displays in map form the most recent RECs that have come into existence by applying the regulations set forth by the most recent directives, with a focus on the cases that have applied for accreditation to the GSE following the promulgation of Law 8/2020.

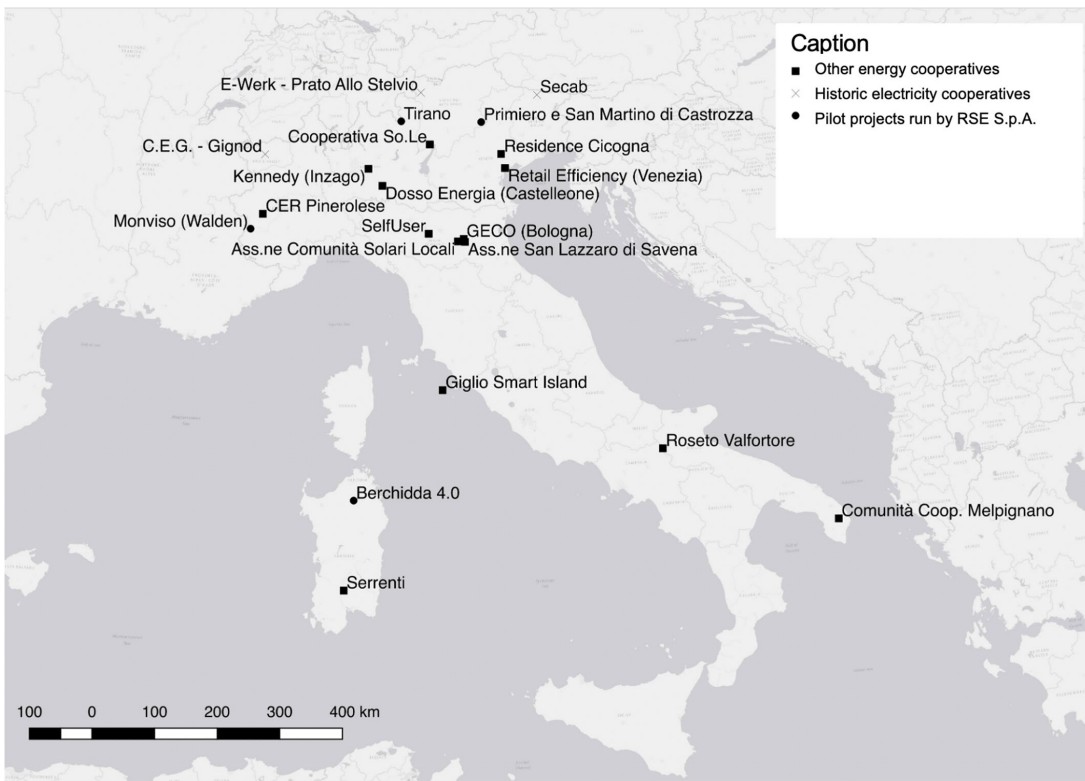

**Figure 1.** Geographical distribution across Italy of the energy enterprises that do not comply with the current legislation. It includes pilot projects from RSE, by also excluding the historical cooperative initiatives from the Alpine regions. Source: © authors.

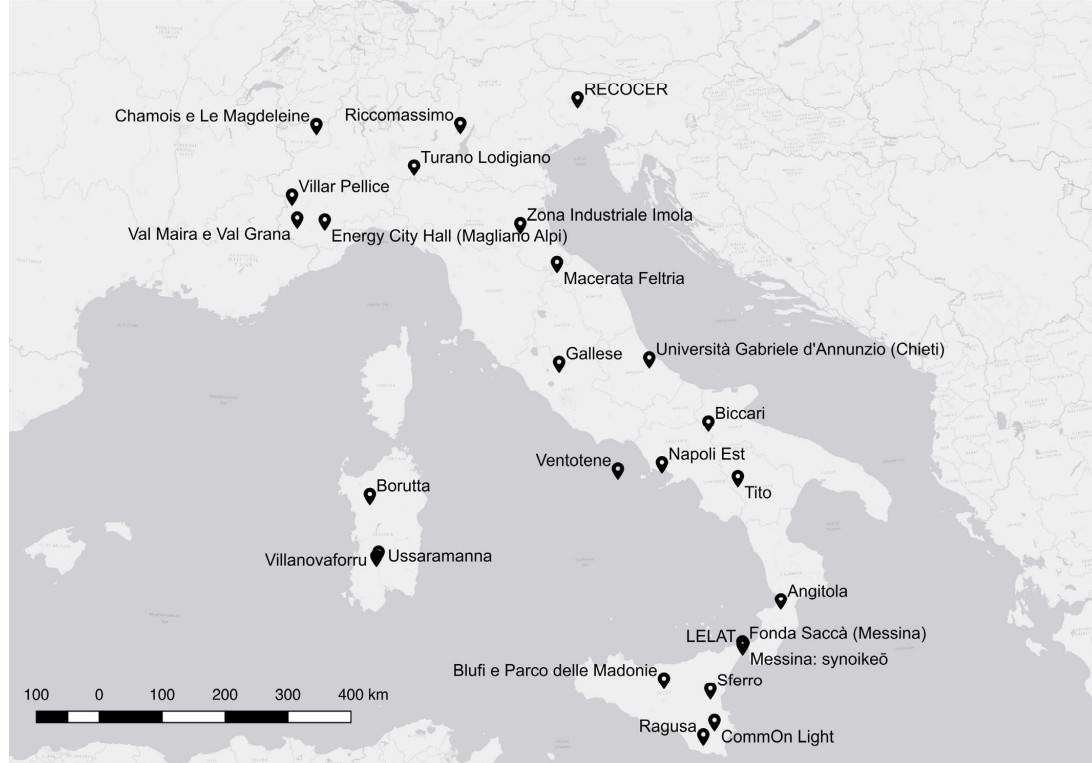

**Figure 2.** Geographical distribution across Italy of the renewable energy communities approved, or in the accreditation process, that comply with National Law 8/2020. Source: © authors, based on RSE's database.

Desk analysis played a knowledge-based role for the following research steps, which delve into the peculiarity of the topic with the aim of sorting and dividing the different types of community projects in a reasoned manner on the one hand, and collecting a series of heterogeneous expert opinions on the other hand. In this sense, for the first purpose, the answers are a clustering phase of community energy projects (Table 1).

**Table 1.** Analytical clustering of the three interpretative models of RECs.

| | CLUSTER 1<br>Public Lead Model | CLUSTER 2<br>Pluralist Model | CLUSTER 3<br>Community Energy Builder Model |
|---|---|---|---|
| **Type of communities and stakeholders** | Local public–private proposers; strong role of the public actor | Application of horizontal community models | Virtual intermediation between local projects and individual consumers |
| **Generatedbenefits** | Public–private partnership to create collective and local benefits | Citizen members and prosumers; coalitions of local actors | Alternative energy consumption models and action on consumer savings |
| **Recruitment and participation processes** | Predominantly top-down process and modus operandi | Predominantly bottom-up process and modus operandi | Heterogeneity of approaches between top-down and bottom-up |

In order to pursue the research objectives described in the Introduction, we decided to identify certain clusters in which to place the case studies, according to the individual features of each, in order to account for the heterogeneity that characterises energy communities. As indicated by Candelise and Ruggeri [39], this heterogeneity is expressed in terms of the actors involved, the diversity of the objectives set by each individual trial, and, above all, the different organisational forms adopted. In this sense, the attempt at modelling underlying the taxonomy is based on a series of key concepts, identified by cross-referencing the reference literature—both Italian and Anglo-Saxon—and the observation of the cases reviewed using desk analysis.

*3.2. Focus Group*

The focus groups held online (in compliance with the restrictions due to the COVID-19 pandemic) involved several experts appropriately divided for each individual meeting. During the first step of research, an initial cognitive meeting was held with some experts, where they discussed with an expert panel of six people who were specialists on the development of energy communities and collective practices from the bottom up on the ecological transition and forms of energy sustainability in Italy. The experts panel comprised three academic professors, the president of an energy cooperative, and three activists involved in the organisation and dissemination of renewable energy projects.

A second set of meetings involved the organisation of three focus groups dedicated to different categories of stakeholders divided according to the basis of specific criteria. With respect to the three thematic focus groups, the first focus group aimed at interacting with members from public administration, with a special interest in local administrators of small municipalities that have activated or are activating an REC. Two mayors, two civil servants, two energy consultants, and three members of a national network of municipalities were involved in the first focus group. The second focus group was dedicated to discussions between private actors and associations from the business world, thanks to the confrontation with 12 individuals who were representative of different entrepreneurial companies and organisations involved in energy transition projects. The third focus group focused its attention on the Third Sector and foundations, by interacting with 11 individuals from bank foundations, Third Sector organisations, and philanthropic foundations involved in different research activities.

The comparison with these three stakeholder categories enabled the collection of further valuable points of view for the following reporting activities, for a complete return of the research activities conducted. A fundamental element for holding the focus groups is the procedure of sharing and comparing [54] different points of view, explicit experiences, reactions, and observations to comments by other participants, so as to allow an ongoing comparison between the actors involved, collecting heterogeneous interpretations and points of view.

*3.3. Introduction to the Case Studies*

The third research method refers to the realisation of a series of case studies, undertaken with qualitative research techniques such as fieldworks and semi-structured interviews with private witnesses, corresponding to local actors who are essential in the cognitive process of each experience taken into consideration. According to Yin [55], case studies facilitate the search for conceptual patterns and categories, which helps to understand a certain phenomenon. Furthermore, as Mabry argues [56], case study research commonly scrutinises not only statistics and numerical data of a case, such as how many people are involved or affected and how indicators of impact vary over time, but is even closer to the experiences and perceptions of participants. To address these aspects, we investigated nine case studies, with an approach that also took inspiration from a threefold subdivision of the conceptual phases behind the development of a case study devised by Fareri [57]: (1) the construction of a timeline of the main significative events in the decision-making processes; (2) the construction and the analysis of actors frameworks, their features, resources, interactions, and positions; (3) the critical interpretation of the decision-making processes in view of the research questions. These three phases were not blindly followed. Rather, they helped to navigate in the cognitive process of gaining information from different case studies.

For the sake of completeness, the nine case studies have been divided as follows, according to the threefold models.

- Case studies for the 'public lead partner model': *CommOn Light* project in Ferla (Sicily); *Kennedy S.r.l.* in Inzago (Metropolitan City of Milan); *Energy City Hall* project in Magliano Alpi (Piedmont).
- Case studies for the 'pluralist model': *Comunità Energetica e Solidale di Napoli Est* in Naples (Campania); *Comunità Energetica Alpina di Tirano*, in Tirano (Lombardy); GECO (Green Energy COmmunity) project in Bologna, Pilastro neighbourhood (Emilia-Romagna).
- Case studies for the model of 'community energy builders' (CEB): project of renewable energy community of Biccari (Puglia), where *ènostra* is identified as CEB; *Condominio agricolo di Ragusa* (Sicily), where *Enel X* is identified as CEB; RECOCER project (Friuli Venezia Giulia), where the Energy Centre from the Polytechnic University of Turin is identified as CEB.

In order to approach the case studies, there was a need to outline an analytical framework capable of systematising the in-depth activities of the case studies through a useful tool for the identification of the main innovations that join the practices reviewed. This exercise entails a reasoning around the key issues introduced with reference to desk analysis and the clustering phases of the three analytical models and, at the same time, a consideration on the governance model undertaken, looking at different dimensions: actors, their positioning and role within the experimentation, objectives and activities planned or already taken and related technologies, and involvement and participation processes, with particular reference to the community that the individual project intends to reach. In addition to these aspects, there are others, deemed necessary to provide a comprehensive framework of the 'voices' to be taken into consideration during the study and research phases. Reference is made here to the regulatory and legislative framework within which the experimentation takes place. According to some recent contributions from the reference literature, it was decided to outline a business model canvas, to be seen as a tool for guidance to observe the main characteristics of the individual case studies, in light of the regulatory reference framework, and the governance models underlying each case. The business model canvas is used to describe and compare the main dimensions and characteristics of the community energy business model examined, derived from the literature that helps to understand pre-existing business models [58]. In this sense, the framework 'per canvas' is also aimed at identifying the challenges that each REC project is called upon to address, proposing alternative solutions to the energy market.

Adapting the business model canvas elaborated by Reis et al. [58], the scheme set out (Figure 3) provides a twofold macro-division of the items to be analysed, distinguishing an input dimension and an output dimension, i.e., separating the organisation of the project from the results, whether they are expected or already achieved. The canvas business model draws on analytical frameworks typical of management studies, which traditionally answer the number of questions concerning the growth of business activities in business-to-consumers (B2C) models. Based on the definition of Osterwalder and Pigneur [59], a canvas business model was used, for the purposes of this research, to accurately describe and compare the main dimensions of business models of energy community projects [58], in order to understand which models are currently established and subject to possible implementation or replication. The distinction between input and output splits among the elements that launched the individual REC project, even on a preliminary basis, and the aspects resulting instead from the actual start-up, implementation, and consolidation phases of the experimentation. For the inputs, the attention turns to five elements: the objectives of the project have different nuances, including multiple elements, from the dispute against energy poverty to collective self-consumption; the actors involved, including stakeholders, public actors, providers of technical expertise, and investors; the resources taken by an economic point of view (monetary investment and costs of project start-up) and a technical point of view (concerning the installation of the systems); the technological aspect, to identify the energy sources taken into consideration, with a strong emphasis on solar energy and the use of photovoltaic systems; the regulatory and policy framework within the project is set.

### COMMUNITY ENERGY MAP - Business Model Canvas
Based upon Reis et. al. (2021)

| INPUTS | | OUTPUTS/OUTCOMES | |
|---|---|---|---|
| **ACTORS**<br>- promoter stakeholder(s)<br>- REC's members<br>- public actor (municipality)<br>- public actor (Region)<br>- technical know-how<br>- external investors<br>- other actors, and the relevant roles | **AIMS AND ACTIVITIES**<br>- energy co-production<br>- energy distribution<br>- self-production and self-consumption<br>- net metering<br>- energy self-sufficiency at a local scale<br>- energy efficiency | **PROCESS**<br>- how the REC is created<br>- how the engagement is enacted (top/down vs bottom/up) | **COMMUNITY**<br>- local community (territory)<br>- intentional community (adhesion to the REC project)<br>- prosuming<br>- territorial cooperation |
| | **TECHNOLOGIES**<br>- energy sources' typology<br>- description of the facilities | **PARTICIPATION**<br>- degree of «openness» (limits and restrictions)<br>- degree of «pluralism» (heterogeneity and diversity of the involved actors)<br>- legal model that regulates the REC: how the participation is regulated<br>- factors influencing the involvement (Soeiro and Ferreira Dias, 2020) | |
| **RESOURCES (support from external actors)**<br>- activation costs<br>- facilities installation<br>- public investments<br>- private investments<br>- outsourcing | **POLICY FRAMEWORK**<br>- normative framework (if prior to Law 8/2020)<br>- typology of policy-framework | **BENEFITS**<br>- economic and individual benefits (deductions in energy bills)<br>- collective benefits for the local community<br>- shared values<br>- revenue stream and other sorts of benefits | |

**Figure 3.** The business model canvas interpretation of the characteristics of the RECs explored in the case studies. Source: Authors' elaboration with a re-adaptation from Reis et al. [58].

For the outputs, we observe elements that become tangible and observable once the project has started: first of all, the process of establishing the REC, to categorise the mechanisms of engagement and involvement of the participants. This involves retracing the dualism between top-down and bottom-up models; the characteristics of the community involved between forms of prosuming and territorial cooperation, to understand whether it is a community configured on a local territorial perimeter, or whether it prefers instead a logic of adherence to the project regardless of territorial proximity, distinguishing itself as an

'intentional' community; participation, degree of pluralism, and the legal model adopted, identifying the factors that influence the participation. In this regard, Soeiro and Ferreira Dias [60] identify four underlying elements of energy community forms: trust, since 'energy decentralisation processes require a good dose of trust among participants' [61]; social standards, useful for regulating forms of cooperation in RECs [62]; community identity, referring to interests that combine collective and individual mobilisation; environmental concerns and the desire to contribute to the process of ecological and energy transition.

The last output dimension concerns the benefits generated by the REC project, with a distribution between individual economic benefits that, in this case, become economic savings for consumers against energy expenses, collective benefits that involve reinvestment in activities or improvements that are useful to the entire population, and forms of revenue stream, broadly understood. The benefits also include a value aspect of sharing decarbonisation goals.

## 4. Three Paradigmatic Examples among the Case Studies

### 4.1. CommOn Light Project

#### 4.1.1. Brief Description and Aims of the Project

CommOn Light (Figure 4) is the first energy community project realised in Sicily, thanks to a collaboration between the Municipality of Ferla (Province of Syracuse) and the University of Catania, within the interdepartmental research project TREPESL (Energy Transition and New Models of Participation and Local Development). The REC is currently materialising through the installation of a small 20 kW photovoltaic system on the roof of the Ferla Municipality, which was realised following a debate with the Superintendence for Cultural Heritage in order to identify a design solution that would allow for the best integration in the landscape. The connection of the plant to the electricity grid is imminent, thus making effective the start of the operation. The project adopts the form of a non-recognised association, which the Municipality of Ferla and four citizens (two individuals and two businesses) are currently members. From a technical–operational point of view, the aim of the project is to achieve a limitation of external energy supply, focusing on on-site storage. This is achieved between the community's ability to produce and store energy, and the community's own energy needs, towards an electrification of consumption. In this way, the Ferla REC intends to provide environmental, economic, and social benefits to its members, with an all-encompassing effort on several objectives such as energy accumulation, storage, production of energy from renewable sources, and allocation of specific resources in the budget for investment in renewable resources, while also nurturing a culture of reference through training activities, participation in conferences, and publishing activities and civic education on environmental sustainability, maintaining an open structure that provides for the inclusion of anyone who requests it. The work team is mainly composed of the two institutional players (the university and the municipality) in a municipality that has already paid some attention to environmental sustainability issues. There is no role for the Region, thus summarising the actors involved in three categories: Ferla Municipality, with regard to political, communication, and engagement competence; the municipality's technical office for the construction of the plant; and the University of Catania for legal–administrative competence.

#### 4.1.2. Economic and Technological Investments

The funding framework for the project is part of the 2014–2020 Sicily ERDF Operational Plan, to which additional resources from municipal 'subthreshold' funds were added to work on the communication profile. The university dealt with the administrative and compliance profiles, while the municipality created the communication campaign to the inhabitants. As the mayor argues, the sums allocated to municipalities in the centre–south for energy efficiency often merge into municipal assets tout-court, without necessarily considering the purposes related to ecological and energy transition. Instead, the sums in the NRRP (the National Plan for the Post-pandemic Recovery, PNRR) will be used

exclusively to finance installations. However, compared with an estimate of EUR 2.2 billion in the NRRP for small municipalities, there is nothing new for small plants such as the one in Ferla, leaving a regulatory void for areas subject to landscape constraints, as in the case of the building on which CommOn Light's photovoltaic plant stands.

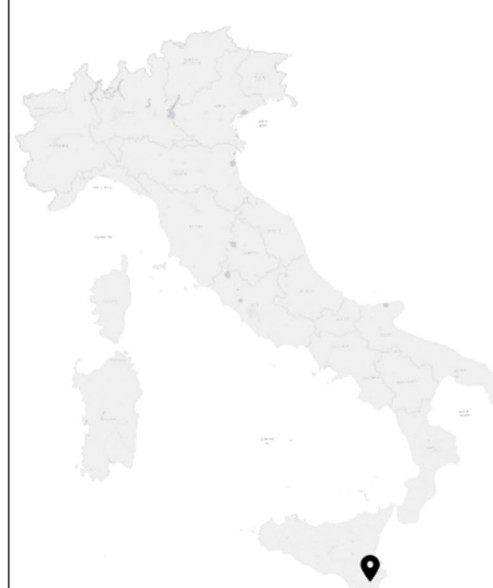

**Project name:** *CommOn Light*
**Location:** Ferla (Sicily)
**Analytical cluster:** public lead model
**Stakeholders involved:**
- Municipality of Ferla
- Technical office of the Municipality of Ferla
- University of Catania
**Local actors surveyed:**
- Marisa Meli (professor, University of Catania)
- Mayor of Ferla
- Milena Pafumi (lawyer, PhD)
- Enrico Giarmanà (Master's at the University of Catania)

**Figure 4.** Information about *CommOn Light* project.

4.1.3. Forms of Management and Community Engagement

The municipal resolutions, when the REC started, took shape from the first acts formulated in March, April, and May 2021. However, the process for the installation of the plant on the Town Hall building dates back to 2017–2018. The Policy Act was enacted in March 2021, followed by the expression of interest for external users interested in the project. In May 2021, the articles of the Association Statute, the articles of the REC 'CommOn Light Association', and the rules of procedure, signed by the members for the distribution of incentives (see section on benefits), were approved. For the implementation of the photovoltaic plant, the municipality had to wait, as mentioned, for a 'go-ahead' from the *Sovrintendenza ai Beni Culturali* (Monuments and Fine Arts Office) with reference to the landscape constraint, following a week of persuading. In this sense, the colouring of the plant in a brown shade allowed for a limited impact on the landscape, net of higher costs and a foreseeable lower yield of the plant itself. For the management of the facilities, the municipality provided the photovoltaic plant available to the community through an open-ended loan agreement, retaining ownership and agreeing on a final term linked to the duration of the REC.

Citizenship involvement practices started in January 2021, in a gradual manner, with a communication campaign launched in March 2021, the day after the publication of the notice of expression of interest on the Ferla Municipality's notice board. On 28 March 2021, the constitution of the energy community was made official to the citizenship. From that moment on, through the social media channels of the municipality, the invitation to participate in a call for interest was formalised, collecting the first adhesions, including heterogeneous profiles of individuals and owners of small–medium enterprises in the retail sector. Around 9–10 PODs answered the expression of interest. Further accessions were temporarily halted due to the constraint on the connection of an REC to the low voltage cabin, in order to avoid overloading.

4.1.4. Benefits

The benefits envisaged occur for an allocation that distinguishes between economic revenue stream and incentive service. For the latter, 20% of the share paid to all members of the community pertains only to the encouragement of adhesion to a renewable energy use project, with a windfall benefit guaranteed to all members. A 30% share is paid out in proportion to the percentage of energy shared by each member, with this percentage intended to act as an incentive to concentrate consumption during daylight hours when the photovoltaic system is at full capacity. The remaining 50% is instead paid to the members who act as producers. So far, it is only the Municipality of Ferla that is the protagonist of the initiative, but this configuration may change as new members join.

For this assumption, there was a fiscal study undertaken on revenue flows. The working group considered it to be consistent with the objectives of the REC to allocate some revenue to a stock for operating costs: the 50% that goes to the producers in power is allocated to a stock, but with two-thirds of the votes of the assembly, this percentage can be allocated to potential new producer–members. To date, the Municipality of Ferla does not collect these resources, but allocates them to other green initiatives or to reinvestment in other plants, thus generating collective benefits in the path towards ecological transition.

*4.2. Comunità Energetica e Solidale di Napoli Est*

4.2.1. Brief Description and Aims of the Project

The Energy and Solidarity Community of East Naples (Figure 5) was one of the first trials to be set up in compliance with Law 8/2020, through the agreement between the number of key local players in the social fabric of the San Giovanni a Teduccio neigh-bourhood and the entire city of Naples. The philanthropic organisation of religious origin and now a foundation under private law, *Fondazione Famiglia di Maria* ('Family of Holy Mary' Foundation), which manages a socio-educational centre in the fragile San Giovanni a Teduccio district and operates in the social services sector with a special focus on minors, and the Fondazione con il Sud, a philanthropic body with a long history of social work in the state capital of Campania, were two of the stakeholders involved. These two founda-tions were joined by *Legambiente Campania*, which provided technical competence on the development of the REC, together with 3E and Italia Solare for the supply of photovoltaic panels, installed on the roof of the socio-educational centre where *Fondazione Famiglia di Maria* operates. According to the actors interviewed, the project lacked the economic and organisational support of local and regional administrations.

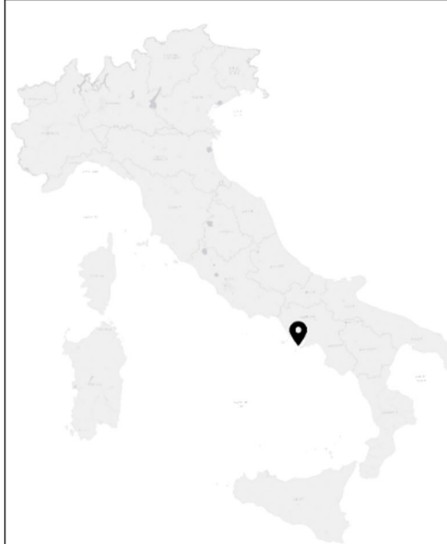

**Project name:**
*Comunità Energetica e Solidale di Napoli Est*
**Location:** Naples (Campania)
**Analytical cluster:** pluralistic model
**Stakeholders involved:**
- Fondazione Famiglia di Maria
- Fondazione con il Sud
- Legambiente Campania
- 3E – Italia Solare

**Local actors surveyed:**
- Felice Petillo (lawyer, General Secretariat of the project)
- Anna Riccardi (President of *Fondazione Famiglia di Maria*)

**Figure 5.** Information about *Comunità Energetica e Solidale di Napoli Est* project.

The deed of incorporation of the Energy and Solidarity Community of East Naples identified a twofold objective, which defined the activities undertaken by the community. From a cultural point of view, the project was aimed at energy education paths, with training activities on ecological transition issues for families subject to socio-economic difficulties, in a neighbourhood with a working-class past torn by the presence of organised crime. Secondly, from an operational point of view, the objective was the production of renewable energy in a logic of sharing, overcoming as much as possible the use of fossil energy and operating as a replacement in terms of supply to homes that have structural conditions that are not easy for energy efficiency solutions. The project envisaged the involvement of households in awareness-raising paths towards sustainable and conscious energy consumption. To pursue these objectives, the energy community was established by installing a photovoltaic system on the roof of the building housing the *Fondazione Famiglia di Maria*.

### 4.2.2. Economic and Technological Investments

The implementation of the photovoltaic system was made possible by a total investment of EUR 100,000 equally distributed between two sources of financing: 50% of the investment was borne by Fondazione con il Sud for social operations, thus including a share between EUR 10,000 and EUR 15,000 for the start-up of training workshops on energy and environmental issues. The remaining 50% of the planned amount was financed by an eco-bonus through an invoice discount and the transfer of credit to the companies involved.

The photovoltaic system has a power output of 55 kW, and it was planned to produce and share the energy produced among 40 families, whose connection points are subtended by the distribution network to which the system is connected under the same MV/LV cabin. *Fondazione Famiglia di Maria*, an organisation widely recognised for its social inclusion and cohesion activities, played a key role in the organisation and dissemination of the project among the social fabric of the district. However, the project was the result of a joint effort by several actors to deal with the above-mentioned legal issues for the creation of the REC.

### 4.2.3. Forms of Management and Community Engagement

The idea of launching this project took shape mainly through in-progress workshops on the meaning and the objective of an 'energy community', together with environmental education courses related to the theme of urban decay. The intertwining with local issues of decay and fragility strongly characterises the social impact of the project, in a context where the development of RECs still appears slow and problematic. In the face of these fragilities, the project responded to the objective of generating 'shared energy' produced from renewable sources, together with the dissemination of a culture of eco-sustainability. The REC was established on 17 March 2021, as an unrecognised association and not a Third Sector body, to pursue non-profit solidarity-based and socially inclusive welfare activities, in accordance with the laws of the European and national regulatory framework. From a legal point of view, the realisation of the REC had to face some technical and urban planning constraints, related to the installation of the photovoltaic plant, as the municipality required the regional environmental authorisation in compliance with landscape constraints. However, the legislation refers the Region to control with authorisation tasks for plants above a certain size, which is not reached by the San Giovanni a Teduccio plant. Moreover, according to point A6 of Presidential Decree 31/2017 on simplified authorisation of the landscape, the installation of photovoltaic panels without the need for a landscape authorisation is possible on flat and non-sloping roofs, as they are not visible from the outside, as in the case of the building housing the *Fondazione Famiglia di Maria*. Nevertheless, despite this evidence, tension arose with the Municipality of Naples over the authorisation of the intervention, which led to a request for reconsideration, and it was only recently accepted after a long procedural requirement.

With regard to social cohesion and the involvement of citizenship, the social entertainment and support activities for families in difficulty played a fundamental role, finding an impact in the collaboration between the *Fondazione Famiglia di Maria*, as promoter, and

the Fondazione con il Sud, as co-financer. The community engagement process saw the involvement of individuals and families who took an interest in the project, activating trust and word-of-mouth mechanisms among the frequenters of the *Fondazione Famiglia di Maria* centre. A final note revisits the temporariness of the project: the Foundation's board of directors expires in 2024, and the positions will need to be renewed. We have set the goal of reaching another 20 families in addition to those already involved; it will be important to continue the work initiated by President Riccardi, in case the Foundation's membership should change.

### 4.2.4. Benefits

The range of benefits emphasises the community-building aspects: before the possible savings in expenditure for families, the participation in an REC implies a personal interest in terms of environmental sustainability, fuelled—in this case—by the innovative path undertaken by the *Fondazione Famiglia di Maria*, which is capable of combining training and educational activities with tangible actions aimed at energy transition, thus going beyond its traditional scope of action. Secondly, a tangible economic benefit for participating families was identified, with an expected annual saving of approximately EUR 300 on each family's energy expenses.

### *4.3. RECOCER Project*
### 4.3.1. Brief Description and Aims of the Project

The RECOCER project (Coordinated Direction of Renewable Energy Community Establishment of Processes in the Territory, 2021–2023) (Figure 6) represented the first case of a strategic multi-annual initiative to establish an REC in the Friuli Venezia Giulia Autonomous region. From an administrative point of view, it involved the *Comunità Collinare del Friuli* (a local authority representing 15 municipalities), while from a technical and scientific point of view, it benefited from the support of the Energy Centre of the Politecnico di Torino. The idea was born with the intention of reproducing, with supra-municipal governance, the approach already tested by the research group of the Polytechnic University of Turin, on the occasion of the REC of Magliano Alpi, inaugurated in December 2020 in Piedmont. The area involved has a population of approximately 50,000 inhabitants and sees an initial context of operation in the Municipality of San Daniele del Friuli, where a school has a 55 kW photovoltaic system.

The *Comunità Collinare del Friuli* launched an energy transition path with prodromal purposes for the entire region. Reading from official documents, the RECOCER project intended to provide an organisational basis capable of implementing energy governance of the territory, through the use of renewable energy sources, with systemic benefits for all participating municipalities. The intention, by 2022, was to make available to all the municipalities of Friuli Venezia Giulia the coordination and management abilities tested with the project of the *Comunità Collinare del Friuli*, which historically plays a role as an interface between the municipalities and the energy supply network, in particular, gas, with the ultimate goal of advocacy and a better price on energy. When initiating the trial, the Energy Centre outlined a business plan for the various development phases, and it built a control and training booth with the *Comunità Collinare*. The objective was to build innovative business models aimed at sustainable local development, also governed by a 'manifesto' for the development of energy communities, developed by the Energy Centre of the Politecnico di Torino to raise awareness of the active support for energy community projects, to catalyse the ability of the various public and private actors (municipalities, universities, companies, citizens), and to build a constant dialogue with national standardisation and regulatory authorities.

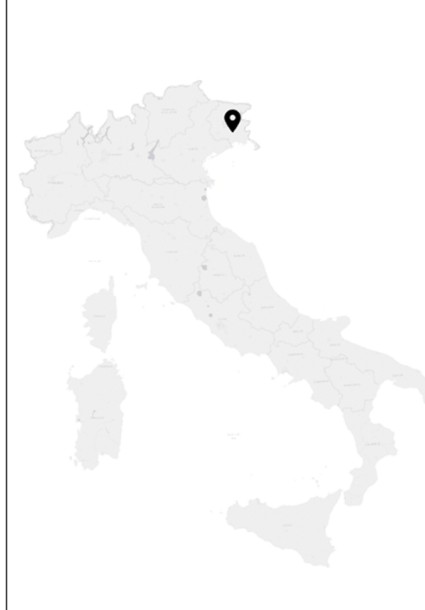

**Project name:**
*RECOCER* project, by the Energy Center at Politecnico di Torino
**Location:** *Comunità Collinare del Friuli* (Friuli Venezia Giulia)
**Analytical cluster:** 'community energy builders' model
**Stakeholders involved:**
- Energy Center at Politecnico di Torino
- 15 Municipalities belonging to the *Comunità Collinare del Friuli* (Buja, Colloredo di Monte Albano, Coseano, Dignano, Fagagna, Flaibano, Forgaria nel Friuli, Majano, Moruzzo, Osoppo, Ragogna, Rive D'Arcano, San Daniele del Friuli, San Vito di Fagagna, and Treppo Grande)
- Municipality of San Daniele del Friuli

**Local actors surveyed:**
- Sergio Olivero (engineer, Energy Center at Politecnico di Torino)
- Romano Borchiellini (coordinator of the Energy Center at Politecnico di Torino; project coordinator)

**Figure 6.** Information about *RECOCER* project.

4.3.2. Economic and Technological Investments

From an economic point of view, funding was allotted to the *Comunità Collinare* by the Autonomous Region of Friuli Venezia Giulia, making available EUR 5,400,000 to be spent by 2023. The RECOCER Project was defined thanks to a feasibility study commissioned by the *Comunità Collinare* to the Energy Centre in the period 2019–2020.

From a technological point of view, as introduced in the first section, the first activated RECOCER fell back into the Municipality of San Daniele del Friuli, legally established on 14 October 2021 with a 55 kW photovoltaic system installed on a school building, playing a prosumer role.

4.3.3. The Role of the 'Community Energy Builder'

The Energy Centre of the Politecnico di Torino plays the role of community energy builder that goes beyond mere technical–scientific consultancy for the start-up of RECs, dealing with the involvement of all territorial stakeholders for the establishment of a sustainable and inclusive governance aimed at the implementation of RECs. The birth of the Energy Centre dates back to 2016, when the Politecnico di Torino launched the Energy Centre Initiative (ECI) in order to start a series of actions and projects of scientific support and to serve as a strategy board to local authorities and national and transnational bodies, on energy policies and technologies to be adopted in a decarbonisation pathway. More generally, the Energy Centre aims to build the national and European networks as an incentive for the development of new business initiatives in the energy sector through the opportunities provided by academic research, innovation, and partnerships. In dialogue with local authorities, it develops business plans, feasibility studies, and support in the transposition of reference directives and reference standards, then it identifies pilot territorial contexts. The project of Friuli took shape to reproduce the working framework already used in the Magliano Alpi experience.

Today, as a founding partner of the Italian Forum on Energy Communities (IFEC), the Energy Centre is collaborating with several municipalities that have signed a collaboration agreement with the Municipality of Magliano Alpi, pursuant to Art. 15 of Law 241/1990, to facilitate the realisation of other energy communities with the same methodological approach and business plans tailored to the specific territorial features.

### 4.3.4. Forms of Management and Community Engagement

Although the first reconnaissance for an REC in Friuli-Venezia Giulia can be traced back to 2018, the first feasibility studies leading to the RECOCER project date back to the summer of 2019, characterised by meetings among local administrations. The Energy Centre's intervention followed, fostered by a work of persuasion and building awareness of the benefits and potential of an REC. The organisational process was started along the lines of the Magliano Alpi experience, then the operational phase began in 2021, thanks to regional funds, with the establishment of the TTC (Community Technical Team) to ensure an effective control in the start-up phases. Supporting the network of municipalities, 'GO-CER', which is a network of energy community operational groups that aim to establish local supply chains for the planning, implementation, and management of RECs, was set up, maximising local added value for sustainable green development. Mentioning this collaborative asset allows us to outline the expected benefits of RECs, although these are still to be read in power.

### 4.3.5. Benefits

RECOCER's official website reports five purposes of the project, coinciding with the expected benefits: (1) the creation of value through innovation in the way of generating, consuming, and managing energy, to be reinvested in the territory; (2) the reduction of electricity bills, to create savings according to the ability to use self-produced energy from renewable sources; (3) the sharing of planning, installation, and management standards of electrical and energy systems and infrastructures, to ensure interoperability among the nodes of the territorial system from the public and private point of view; (4) the development of local supply chains through synergy among builders, installers, maintainers, and planners to provide services to citizens, with job creation and stimulation towards new economies in the post-pandemic phase; (5) the coordinated procurement of goods and services, creating economies of scale to foster the good practices and experiences, avoiding the multiplication of costs. In light of these objectives, the Energy Centre of the Politecnico di Torino aims to systematise a technical–scientific approach that gives long-term solidity to business models.

## 5. Discussion: The Elements of Organisational Innovation

### 5.1. The Combination of Different Know-Hows between Public Administration and Other Institutions

The development of RECs requires a number of preconditions in terms of skills, resources, and capabilities, created with ad hoc solutions according to the specificities of local contexts. With regard to this aspect, two main obstacles stand out: on the one hand, the search for the economic sustainability of these initiatives, and on the other hand, the inclusiveness in terms of accessibility of these initiatives within different territories.

According to cases such as RECOCER and Ferla's REC, the role played by public actors in the co-production of initiatives has certainly proved to be a fundamental factor of organisational innovation. The enlarged governance with the technical skills and knowledge of the territory brought by the universities has become a key factor for the sustainability of the initiatives.

As can be seen in all the cases analysed, the managerial dynamics of initiatives are key factors to make links operational with the territory and communities of practice. In this respect, projects similar to the one led by the Municipality of Biccari were realised thanks to the connections with organisations operating on a national scale, borrowing knowledge among ecosystems and networks of national and international organisations that already conduct advocacy activities for the energy community sector, as well as for the rights that community-based enterprises have in the energy transition, as in the case of the partnership between 'Friends of the Earth Europe' and 'REScoop.eu'.

### 5.2. The Capacity to Deliver Social and Environmental Impacts

In many of the cases, RECs are run almost as social enterprises, by explicitly declaring the aim of promoting inclusion, cohesion, and of countering situations of marginality, thus using the energy production from RECs and the relevant economic benefits in an instrumental way. The topic of combating energy poverty, for instance, was constantly mentioned both in focus groups (particularly the one involving foundations and Third Sector organisations) and in practical cases. This occurred despite the fact that in the Italian context there is currently no clear reference framework for measuring the social impact of RECs, and no organisations of significant size that are able to systematically adopt these approaches inspired by the principles of social enterprises.

This aspect has been a key factor in the organisational innovation, as it enables not only to address issues related to energy poverty and vulnerability, but also to focus on the implications in terms of inclusion and social cohesion. This issue deserves further reflection with reference to the 'social impacts' of the initiatives and the 'quantity' of social outputs (and outcomes) that the REC initiatives produce. Such a challenge entails the identification of parametric values and the assessment of possible 'substitution effects' with the cost of outputs produced by other economic, social, and energy policies, promoted both locally and nationally. This is pivotal when incentives are to be used to reward initiatives that lead to positive community benefits at a local and community level.

### 5.3. Proximity and Place-Based Approaches as an Added Value of RECs

The use of models highly linked to local dynamics makes RECs a device for the creation of value in terms of 'territorial proximity' [63]. This dimension emerges from the observation of the development process, which results in a complex combination of resources and partnerships that determine its implementation. Given the difficulty of 'small' local actors in the personal development of these projects, the initiatives are developed through the involvement of actors such as public administration, associations, and external community energy builders that facilitate the action of local communities. The territorial dimension of the interaction among a community of users/investors, local actors, and technologies is an essential factor in the exchange of both tangible (financial resources and physical assets, for instance, roof surfaces) and intangible (trust, share capital, contextual knowledge) assets. Territorial proximity is a peculiar feature and reveals the need to employ place-based approaches for the development of such initiatives [64].

According to this aspect, it becomes very difficult to imagine a production of value generated by RECs that relies on virtual community relations, such as peer-to-peer exchange platforms among various consumption and production units, where the latter are solely in the hands of individual actors. Such a scenario would reduce the positive spill-over effects on the local economies, related to the opportunities to generate income and employment in the territories where RECs are launched. Against this backdrop, RECs assume greater relevance when they foster local territorial resources. If we look at the cases by imagining a gradual spread of these initiatives, it seems likely that a number of pressures and tensions will arise with regard to the business model and return on investment. On the one hand, there will be initiatives with a place-based approach, capable of enhancing local resources and generating outputs and outcomes for communities. On the other hand, industrial-type investments will optimise resources to enable the participation of non-local actors that hold certain skills that would otherwise not be activated. If we look at decarbonisation objectives at a national level, the development of RECs will benefit from both approaches. If the former appears at first glance to be more related to the local community engagement, the latter appears to be more connected to the issue of suitable areas within which to place larger RECs' plants. In order to maximise the value elements of both approaches, the regions and local authorities will play a crucial role in appropriately declining the development of the initiatives, so as to maximise the benefits and to limit the possible extraction of value (and increase of conflict) in the territories.

*5.4. Conclusive Comments on Policy Recommendations*

To sum up, two further reflections entail recommendations for the policy-making to support RECs' development and implementation by combining localised engagement processes with the economic sustainability. Against this backdrop, policy recommendations are based upon the threefold subdivision of the organisational models that have been illustrated. RECs represent pivotal initiatives to innovate the energy systems towards the development of community-led decentralised models, and also to make the increasingly inhabited urban environments healthier [65].

The first question concerning the analysis of the specific examples highlights the importance of promoting new approaches for potential policy-makers, entrepreneurs, and individuals capable of developing and activating RECs, with a particular reference to two specific competences:

- Systematising the involvement of the individuals in the entrepreneurial initiative, with reference to the ability to conclude contracts to share benefits and responsibilities among the members of a local community, and to manage the priorities and the interests of a plurality of individuals taking into account the effective participation and representation of needs and expectations in decision-making processes.
- Ensuring a sustainable investment model in REC projects, to mobilise the network of the relations among local actors [66], and also to promote investments related to horizontal subsidiarity or entrepreneurial participation of citizens in the planning of services and spaces for local communities [67].

The second reflection is related to the role that RECs will be able to play in the broader context of ecological transition. Through the implementation of the NRRP guidelines and in the broader framework of regional energy policies (regional climate plans of several Italian regions and allocation of ERDF funds for the promotion of RECs consistent with regional development hypotheses), national policies, within the update of the National Integrated Energy and Climate Plan–PNIEC, are supposed to implement the European Green Deal, by also integrating the legislation of various European plans and programmes such as 'InvestEU', the Connecting Europe Facility (CEF), the European Regional Development Fund (ERDF), the Cohesion Fund (CF), and the Just Transition Fund (JTF). The combination of these two reflections calls for further investigations on the multi-level perspective that lies behind the development of RECs, and the need to increase a broad social acceptance of community energy initiatives.

## 6. Conclusions

The discussion proposed in the previous section enabled us to answer the two research questions posed in this paper, related to the organisational models of renewable energy community initiatives in Italy. In this paper, we attempted to systemise the differentiation that runs between the ways in which RECs are organised and implemented. The main outcomes of the applied research *Community Energy Map*, conducted in Italy, have been illustrated and discussed through a thematization of three organisational models to develop RECs, and an overview of six punctual case studies among the most recently launched initiatives.

To conclude, reflections on further research pathways are to be mentioned, as new issues are arising. Both scholars and practitioners suggest monitoring the RECs' implementation by also involving the social dimension and the local effects generated from a renewable energy community, through the elaboration of evaluation tools for energy transition practices that intend to go beyond the traditional approaches that lead exclusively to economic and financial dynamics. Place-based approaches act as a leverage for a 'fair' energy transition in this regard. Yet, a knowledge gap affects the connection between these different aspects. With this in mind, the transition practices (involving not only RECs, but also collectively owned renewable energy plants, self-consumption schemes, community-based energy efficiency projects, and energy poverty alleviation) can become the vehicle of territorial policies capable of supporting processes of social cohesion and

inclusion, where the energy dimension can trigger local economic developments. In more detail, the challenges to be tackled with respect to a better definition of the impacts of energy transition practices relate to the following dimensions:

- The spatial and environmental dimension: Identifying the scale and nature of environmental impacts through the identification of results (outcomes) produced at micro, meso, and macro scales as variables influenced by the strategic characteristics with which the initiatives were implemented. It involves identifying the social dimension of projects in considering both local energy needs and the different capabilities and endowments that marginalised subjects and territories have in implementing 'fair transition' projects.

- The governance dimension of these practices: With respect to this aspect, energy communities represent a framework of examples of initiatives that are particularly attentive to the issue of fair transition, where it is, however, crucial to observe the role played by policies (plans, partnerships, energy policy) organisations and stakeholders with different interests that consequently direct the mobilisation of both financial and intangible resources.

If new initiatives succeed in working on common frameworks for measuring social outputs, they will also be able to use financial solutions with the declared objective of producing a measurable social benefit while generating a financial return. In this regard, the purposive quality on the part of investors to direct their capital towards initiatives will be pivotal, even to identify the territorial specific social needs that fulfil these impact functions. With respect to the issue of skills, which is crucial to a fair transition mechanism, the gap to be bridged in human capital retraining is a particularly critical issue.

First and foremost, it is necessary to strengthen the internal competencies of public administration alongside financial support measures. Many of the municipalities targeted for funding under the NRRP are unlikely to be able to cope with the needs required in terms of planning competences, with the risk that the organisations outside of local dynamics will promote 'prefabricated models' that could lead to an organisational isomorphism [68], without the real involvement of local communities, without real spin-offs in terms of positive externalities, and with technological solutions incapable of producing added value in the local economies.

A key issue is therefore the possibility to develop valuable conditions of organisational/financial sustainability for projects that necessarily need to take on investors (more or less institutional), and at the same time to develop collaborations among different private actors (e.g., energy service companies, ESCOs) or public actors (e.g., public administration, public housing agencies, group of municipalities). To this end, cooperatives, foundations, and community-based (i.e., collectively owned) capital enterprises emerge as applicable models to support the potential of these local innovations where the required skills include governance planning, partnerships, and community engagement.

Such reflections acquire even more importance in the Italian context if we look at the need to make the most of the potential offered by the recent overall transposition of the RED II Directive through Legislative Decree 199/2021. The enlargement of the perimeter on which RECs can insist on the primary booth and the increase in the incentivised size of plants to 1 MW should be exploited to move from small plants, promoted by the early adopters of these initiatives, to a main-stream growth of the REC model on the national territory. In addition to the possibility of involving wider communities, the regulations work to promote a progressive increase in the complexity of RECs, which in coming years will be engaged in the thermal sector, in shifting loads towards less polluting vectors (starting from the possible electrification of mobility and heating systems, where applicable and convenient), in participation in energy markets such as the flexibility one, and in the definition of energy policies that are strongly coherent with local development policies.

**Supplementary Materials:** The following supporting information can be downloaded at: https: //www.mdpi.com/article/10.3390/su15031997/s1.

**Author Contributions:** Conceptualization, L.D.V., L.T. and M.Z.; Methodology, L.T.; Validation, M.Z.; Formal analysis, L.D.V.; Investigation, L.D.V. and L.T.; Resources, M.Z.; Data curation, L.D.V.; Writing—original draft, L.D.V.; Writing—review & editing, L.D.V., L.T. and M.Z.; Supervision, M.Z.; Project administration, L.T.; Funding acquisition, M.Z. All authors have read and agreed to the published version of the manuscript.

**Funding:** This work has been financed by the Research Fund for the Italian Electrical System under the Contract Agreement between RSE S.p.A. and the Ministry of Economic Development—General Directorate for the Electricity Market, Renewable Energy and Energy Efficiency, Nuclear Energy in compliance with the Decree of 16 April 2018.

**Institutional Review Board Statement:** Not applicable.

**Informed Consent Statement:** Informed consent was obtained from all subjects involved in the study.

**Data Availability Statement:** No new data were created for this paper.

**Conflicts of Interest:** The authors declare that they have no known competing financial interests or personal relationships that could have appeared to influence the work reported in this paper.

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
