# Peer review of "How Can We Frame Energy Communities’ Organisational Models? Insights from the Research ‘Community Energy Map’ in the Italian Context"

_sustainability, doi:10.3390/su15031997_

Round 1
Reviewer 1 Report
Dear Authors,
In my opinion, the presented work is well structured and focused on a actual "problem" which seems to be present in more and more countries, not only in European area. It is to mention that solutions in the scenario of switching from conventional fossil fuel energy to renewable energy has many variable depending to countries and regions, depending to local and national laws, energy networks and so on. After carefully reading the submitted work I consider it worth to be accepted in present form.
review of the available literature regarding the dynamics of pressure evolution during gaseous ethane-air mixture explosions in enclosures is very well written and comprehensible. The results are clear and very well summarised. In my opinion, the topic of the paper is not of the highest interest but could be important to specific audience both in academia and the industry. Considering the stated I would recommend it for publication in present form.
Author Response
Dear Reviewer,
Thanks for reviewing our manuscript and for the positive feedback.
With best wishes
Reviewer 2 Report
Thanks a lot for the opportunity to read this very interesting paper. I really appreciated the idea and I do think that it provides useful practical insights for the Academia and the energy business community. Here as follows my comments to improve the article.
When on page 2 you refer to the role of ESG factor for decision maker, companies and investors, you should provide citations in support of this.
Regarding decentralization and territorialization you should explain the relevance of focusing your work on local contexts. The relevance of “local” sheds light on the way communities shape places. Here comes into play to investigate how all business happens in such places. You should follow this common thread to stress the attention on the relevance of your topic.
For a more detailed discussion of energy communities, you could mention the following articles:
Gjorgievski, V. Z., Cundeva, S., & Georghiou, G. E. (2021). Social arrangements, technical designs and impacts of energy communities: A review. Renewable Energy, 169, 1138-1156.
Bartolini, A., Carducci, F., Muñoz, C. B., & Comodi, G. (2020). Energy storage and multi energy systems in local energy communities with high renewable energy penetration. Renewable Energy, 159, 595-609.
Moreover, you could mention some papers focusing on the drivers of investments in the energy sector, such as for example
Garau, G., Lecca, P., & Mandras, G. (2013). The impact of population ageing on energy use: Evidence from Italy. Economic Modelling, 35, 970-980.
I would not say “the topic is not new in the Italian context”. Rather I would highlight the need for new local interactions that emerges in the energy communities.
Additionally, you should explain why Italy is a proper field of study for your analysis, making a very brief description of the Italian context, also in terms of firms and individuals that belong to the communities. With this regard you could cite two papers that are important to show the role of SMEs (which are often individual business or family firms and as such strongly connected with the local setting) in the energy sector.
Cariola, A., Fasano, F., La Rocca, M., & Skatova, E. (2020). Environmental sustainability policies and the value of debt in EU SMEs: Empirical evidence from the energy sector. Journal of Cleaner Production, 275, 123133.
La Rocca, T., La Rocca, M., Fasano, F., & Cariola, A. (2022). Does a country's environmental policy affect the value of small and medium sized enterprises liquidity in the energy sector?. Corporate Social Responsibility and Environmental Management.
Please provide definition of “European energy market”.
Liberalization and decentralization are two phenomenon indicating that the local context could matter less in the near future (this apply to many sectors and not only to the energy one): therefore, again you need to convince the reader about the relevance of examining a local environment.
Regarding the “organizational lenses” at the core of your first question, you could mention in the introduction section the main benefits and potential cases of application also in light of the clusters emerging from Table 1.
I find the end of Paragraph 2 too long. Similarly, in Paragraph 3 there is no need of such a large explanation about the characteristics of a qualitative research. In these sections and, more in general, throughout the paper I suggest to be more concise and get the point. Maybe you could reduce the discussion of the three examples or move one or two to the appendix.
In the focus group, how many professors, experts and members from representative bodies and organization where there? How many members for each category? What is the main output of the focus group?
In the business model canvas elaborated, what is the proportion of results expected and achieved put in the output dimension?
Author Response
Dear Reviewer,
Thanks for reviewing our manuscript. The response to your comments is provided below, whereas a revised version of the whole manuscript is attached. Amendments and new parts are indicated in red throughout the attached document.
With best wishes.
REVIEWER 2
|
COMMENTS FROM THE REVIEWER |
RESPONSE TO THE REVIEWERS FROM THE AUTHORS |
|
When on page 2 you refer to the role of ESG factor for decision maker, companies and investors, you should provide citations in support of this. |
We added the following reference: Kaiser, L. (2020). ESG integration: value, growth and momentum. Journal of Asset Management, 21(1), 32-51. |
|
Regarding decentralization and territorialization you should explain the relevance of focusing your work on local contexts. The relevance of “local” sheds light on the way communities shape places. Here comes into play to investigate how all business happens in such places. You should follow this common thread to stress the attention on the relevance of your topic. |
The relevant sentence has been revised as follows, by adding one of the suggested references (i.e., Bartolini et. al., 2020):
“Decentralisation and territorialisation of energy production are the main features on which energy communities are based, insofar as the localisation of energy production towards the aim of making energy production and consumption closer to each other is a pillar of energy community initiatives (1) |
|
For a more detailed discussion of energy communities, you could mention the following articles:
Gjorgievski, V. Z., Cundeva, S., & Georghiou, G. E. (2021). Social arrangements, technical designs and impacts of energy communities: A review. Renewable Energy, 169, 1138-1156.
Bartolini, A., Carducci, F., Muñoz, C. B., & Comodi, G. (2020). Energy storage and multi energy systems in local energy communities with high renewable energy penetration. Renewable Energy, 159, 595-609.
Moreover, you could mention some papers focusing on the drivers of investments in the energy sector, such as for example:
Garau, G., Lecca, P., & Mandras, G. (2013). The impact of population ageing on energy use: Evidence from Italy. Economic Modelling, 35, 970-980. |
The reference Bartolini et. al. (2020) has been added. Please, see the former response, above.
The refrences to Gjorgjevski et al (2021) has been added in the following reframe of the presentation of the two research questions.
“From the outcomes of these prodromal experiences, that suggest a complex combination between social arrangements and technical issues for the development RECs initiatives (5), to research questions arise, by involving an analysis of both organisational models and social impacts of a REC, by also relating to forms of citizen involvement for the development of energy communities:”
Reference 5 is Gjorgjevski et al.
Unfortunately, we find some difficulties in adding the reference to Garau et. al. |
|
I would not say “the topic is not new in the Italian context”. Rather I would highlight the need for new local interactions that emerges in the energy communities. |
The sentence and the relevant part have been modified as follows:
“Localized forms and initiatives of energy communities have begun to appear in Italy in the Alpine areas at the end of the 1800s, stemming from the valorisation of local resources to satisfy the emerging energy needs of the local communities” |
|
Additionally, you should explain why Italy is a proper field of study for your analysis, making a very brief description of the Italian context, also in terms of firms and individuals that belong to the communities. With this regard you could cite two papers that are important to show the role of SMEs (which are often individual business or family firms and as such strongly connected with the local setting) in the energy sector.
Cariola, A., Fasano, F., La Rocca, M., & Skatova, E. (2020). Environmental sustainability policies and the value of debt in EU SMEs: Empirical evidence from the energy sector. Journal of Cleaner Production, 275, 123133.
La Rocca, T., La Rocca, M., Fasano, F., & Cariola, A. (2022). Does a country's environmental policy affect the value of small and medium sized enterprises liquidity in the energy sector?. Corporate Social Responsibility and Environmental Management. |
The research has not been developed to include any reasoning about the Italian SMEs sector. Even though we acknowledge the relevance of this aspect, for the sake of brevity and to avoid the risk to push the analysis out of its research context, we decided not to include the references and not to introduce the suggested topic. We hope you can get our points in this regard. |
|
Please provide definition of “European energy market” |
We have changed the sentence in the following way: The socio-technical organisational concept of Energy Communities has been incorporated by a recent development in the European energy market regulation concerning the relationship between market and consumers ref: https://energy.ec.europa.eu/topics/markets-and-consumers_en |
|
Liberalization and decentralization are two phenomenon indicating that the local context could matter less in the near future (this apply to many sectors and not only to the energy one): therefore, again you need to convince the reader about the relevance of examining a local environment.
|
We didn’t get the reviewer point. We actually do not see this correlation. We reframed the part on progressive liberalisation by adding a reference, and reframing a sentence as follows:
Within a context of progressive liberalisation of the energy market and decentralisation of energy generation activities, in recent years the prominence of end customers has been gaining in importance, resulting in what Goldthau has defined a new polycentric governance of the energy infrastructures
The reference is the following. Please see at the key reference developed by Goldthau: Goldthau, A. (2014). Rethinking the governance of energy infrastructure: Scale, decentralization and polycentrism. Energy Research & Social Science, 1, 134-140. |
|
Regarding the “organizational lenses” at the core of your first question, you could mention in the introduction section the main benefits and potential cases of application also in light of the clusters emerging from Table 1. |
To address this comment, we added the following sentence in the illustration of the reasons that lie behind the first research questions:
“The identification of three ‘organizational clusters’ enable to distinguish three different models about the ways in which community energy initiatives have been framed and developed in the Italian context.” |
|
I find the end of Paragraph 2 too long. Similarly, in Paragraph 3 there is no need of such a large explanation about the characteristics of a qualitative research. In these sections and, more in general, throughout the paper I suggest to be more concise and get the point. Maybe you could reduce the discussion of the three examples or move one or two to the appendix. |
Thanks for this comment. We consider the presentation of the Italian state of play as a necessary part of our paper. We therefore did not reduce section 2. However, we reduced section 3 on methodology, by removing the following parts:
“However, some caveats related to this methodological approach are to be considered. Flyvbjerg (49) points out that qualitative tools ostensibly allow more room for the researcher’s subjective and arbitrary judgment than other methods, as they are less stringent than quantitative analyses, and based on hypothetical and deductive methods. However – Flyvbjerg maintains (49) – experienced case researches cannot help but see the critique as demonstrating a lack of knowledge of what is involved in case-study research. As the development of energy community initiatives according to a national and supranational legislative framework are far from a new topic in the pathway towards energy transition in Italy”
Furthermore, after briefly presenting the significance of qualitative research at the beginning of section 3, we reframed a sentence as follows: “On this basis, and according to the research aims of this paper, qualitative methods enable to add knowledge on the timely topic of RECs development within the energy transition, and to lay down some general key aspects derived from the number of case studies.” |
|
In the focus group, how many professors, experts and members from representative bodies and organization where there? How many members for each category? What is the main output of the focus group? |
We added the requested info, as follows:
“The experts panel was composed by three academic professors, the President of an energy cooperative, and three activists involved in the organization and dissemination of renewable energy projects.”
“Two Mayors, two civil servants for public bodies, two energy consultants, and three members of a national network of Municipalities have been involved in the first focus group.”
“The second focus group was dedicated to discussions between private actors and associations from the business world, thanks to the confrontation with twelve individuals representative of different entrepreneurial companies and organizations involved in energy transition projects.”
“The third focus group focused its attention on the Third Sector and foundations, by interacting with eleven individuals from bank foundations, Third Sector’s organizations, and philanthropic foundations involved in different research activities.” |
|
In the business model canvas elaborated, what is the proportion of results expected and achieved put in the output dimension? |
To our research purposes, the elaborated business model canvas merely has an organizational analysis objective. |
Reviewer 3 Report
This work reported the operational models and organisational frameworks of renewable energy communities, aimed to investigate the social and local implications generated by these community-led initiatives. This work is useful for the energy transition from fossil fuels to renewable sources. Some comments are as follows:
1. The concept and description, togetherwith the language should be improved. The present manuscript are difficult to understand.
2. The author should present some literature carrying out the similar research work, which will be useful to illustrate the significance of this work.
3. The Conclusion is too long. It would be better that the results or important suggestion are placed in Discussion section.
Author Response
Dear Reviewer,
Thanks for reviewing our manuscript. The response to your comments is provided below, whereas a revised version of the whole manuscript is attached. Amendments and new parts are indicated in red throughout the attached document.
With best wishes.
|
COMMENTS FROM THE REVIEWER |
RESPONSE TO THE REVIEWERS FROM THE AUTHORS |
|
The concept and description, together with the language should be improved. The present manuscript are difficult to understand. |
We actually have difficulties to understand this point. What is unclear about the concept and description? What should be improved? |
|
The author should present some literature carrying out the similar research work, which will be useful to illustrate the significance of this work. |
We added a number of references to strengthen the research framework. The new references are the following:
In the introduction: 1. Kaiser, L. (2020). ESG integration: value, growth and momentum. Journal of Asset Management, 21(1), 32-51. 2. Bartolini A, Carducci F, Muñoz CB, Comodi G. Energy storage and multi energy systems in local energy communities with high renewable energy penetration. Renewable Energy [Internet]. 2020 Oct [cited 2023 Jan 13];159:595–609. 6. Gjorgievski VZ, Cundeva S, Georghiou GE. Social arrangements, technical designs and impacts of energy communities: A review. Renewable Energy [Internet]. 2021 May [cited 2022 Mar 9];169:1138–56.
About the liberalization of the European energy market: 48. Goldthau, A. (2014). Rethinking the governance of energy infrastructure: Scale, decentralization and polycentrism. Energy Research & Social Science, 1, 134-140. |
|
The Conclusion is too long. It would be better that the results or important suggestion are placed in Discussion section. |
Conclusion has been reduced and fragmented. The first final reflections that were previously placed in the conclusion, have been replaced to the discussion, in a fourth, concise sub-section titled “5.4 Conclusive comments on policy recommendations”. Conclusions have been revised accordingly. |
Reviewer 4 Report
The review comments for the manuscript, 'How can we frame energy communities' organizational models? Insights from the research ‘Community Energy Map’ in the Italian context', are given below,
1. The work is impressive and presented well. Still, few changes need to be done for betterment.
2. Add the organization of the manuscript at the Introduction section.
3. References should not be grouped like, [11]-[15] or [22]-[25]. Each and every reference must be properly justified and cited.
4. English must be improved with the help of a language expert.
5. Also, Figure 1 context must be changed to English language for better understanding to the readers.
6. Typos and grammatical errors are throughout the manuscript. Kindly check it once.
7. Table should be given proper format.
8. Abstract and conclusion (especially) must be given brief and precise.
Author Response
Dear Reviewer,
Thanks for reviewing our manuscript. The response to your comments is provided below, whereas a revised version of the whole manuscript is attached. Amendments and new parts are indicated in red throughout the attached document.
With best wishes.
|
COMMENTS FROM THE REVIEWER |
RESPONSE TO THE REVIEWERS FROM THE AUTHORS |
|
Add the organization of the manuscript at the Introduction section. |
The organization of the manuscript is actually already indicated at the end of the introduction. From “In order to answer these questions and pursue the aforementioned objectives, the article is divided into different sections…” to the end of the Introduction. |
|
References should not be grouped like, [11]-[15] or [22]-[25]. Each and every reference must be properly justified and cited. |
With particular reference to those quotations, we gathered them in a single “parenthesis” simply to account for the richness of existing studies on the following topics: co-creation of energy projects (references from 11 to 15, now modified from 12 to 16) and the relevance of RECs project amongst the decision makers (references from 22 to 25, now updated from 23 to 26). For the sake of brevity, we gathered these references, insofar as a justification of each reference would fragment the reading. We hope you can understand our point in this matter. |
|
Also, Figure 1 context must be changed to English language for better understanding to the readers. |
Thanks. Figure 1 has been modified accordingly. |
|
Typos and grammatical errors are throughout the manuscript. Kindly check it once. |
Grammar errors and typos have been checked again. Thanks for notifying. |
|
Table should be given proper format. |
We provided Table 1 in a word format, attached to the submission’s files. |
|
Abstract and conclusion (especially) must be given brief and precise. |
The abstract has been revised as follows:
According to the early transposition of the EU Directives by the Italian government, the paper presents some of the outcomes of the qualitative-led applied research titled Community Energy Map, aimed at identifying the main operational models and organisational frameworks put in place for the development of Renewable Energy Communities (RECs). In this respect, the article discusses a threefold subdivision of organizational models to implement RECs: public lead, pluralist and community energy builders’ model. Furthermore, the paper illustrates in details three of the nine case studies dedicated to recently launched RECs, conducted through qualitative fieldworks, to investigate the social and local implications generated by these community-led initiatives. The article stresses the relevance of both the local scale and community-led initiatives in the pathway towards a fair and just energy transition, by discussing how RECs defines new organizational models of distributed energy systems. |